# Formononetin inhibits lipopolysaccharide-induced release of high mobility group box 1 by upregulating SIRT1 in a PPARδ-dependent manner

Jung Seok Hwang[1], Eun Sil Kang[1], Sung Gu Han[1], Dae-Seog Lim[2], Kyung Shin Paek[3], Chi-Ho Lee[1] and Han Geuk Seo[1]

[1] Department of Food Science and Biotechnology of Animal Products, Sanghuh College of Life Sciences, Konkuk University, Seoul, Korea
[2] Department of Biotechnology, CHA University, Seongnam, Korea
[3] Department of Nursing, Semyung University, Jecheon, Korea

## ABSTRACT

**Background:** The release of high mobility group box 1 (HMGB1) induced by inflammatory signals acts as a cellular alarmin to trigger a chain of inflammatory responses. Although the inflammatory actions of HMGB1 are well studied, less is known about the therapeutic agents that can impede its release. This study investigated whether the isoflavonoid formononetin can modulate HMGB1 release in cellular inflammatory responses.
**Methods:** RAW264.7 murine macrophages were exposed to lipopolysaccharide (LPS) in the presence or absence of formononetin. The levels of HMGB1 release, sirtuin 1 (SIRT1) expression, and HMGB1 acetylation were analyzed by immunoblotting and real-time polymerase chain reaction. The effects of resveratrol and sirtinol, an activator and inhibitor of SIRT1, respectively, on LPS-induced HMGB1 release were also evaluated.
**Results:** Formononetin modulated cellular inflammatory responses by suppressing the release of HMGB1 by macrophages exposed to LPS. In RAW264.7 cells, formononetin significantly attenuated LPS-induced release of HMGB1 into the extracellular environment, which was accompanied by a reduction in its translocation from the nucleus to the cytoplasm. In addition, formononetin significantly induced mRNA and protein expression of SIRT1 in a peroxisome proliferator-activated receptor δ (PPARδ)-dependent manner. These effects of formononetin were dramatically attenuated in cells treated with small interfering RNA (siRNA) against PPARδ or with GSK0660, a specific inhibitor of PPARδ, indicating that PPARδ is involved in formononetin-mediated SIRT1 expression. In line with these effects, formononetin-mediated inhibition of HMGB1 release in LPS-treated cells was reversed by treatment with SIRT1-targeting siRNA or sirtinol, a SIRT1 inhibitor. By contrast, resveratrol, a SIRT1 activator, further potentiated the inhibitory effect of formononetin on LPS-induced HMGB1 release, revealing a possible mechanism by which formononetin regulates HMGB1 release through SIRT1. Furthermore, modulation of SIRT1 expression by transfection of SIRT1- or PPARδ-targeting siRNA significantly counteracted the inhibitory effects of formononetin on LPS-induced HMGB1 acetylation, which was responsible for HMGB1 release.

Corresponding author
Han Geuk Seo, hgseo@konkuk.ac.kr

**Discussion:** This study shows for the first time that formononetin inhibits HMGB1 release by decreasing HMGB1 acetylation via upregulating SIRT1 in a PPARδ-dependent manner. Formononetin consequently exhibits anti-inflammatory activity. Identification of agents, such as formononetin, which can block HMGB1 release, may help to treat inflammation-related disorders.

## INTRODUCTION

High mobility group box 1 (HMGB1), a non-histone DNA-binding protein, is a well-conserved nuclear protein that has multiple functions depending on its cellular location. In the nucleus, HMGB1 plays roles in DNA replication, transcription, recombination, and maintenance of chromosome stability (*Stros, 2010*). However, when released by stressed cells, HMGB1 plays a critical role in the inflammatory response and is a late proinflammatory marker in many diseases including sepsis (*Andersson & Harris, 2010*; *Abdulahad et al., 2010*; *Sims et al., 2010*; *Stros, 2010*; *Zhang et al., 2009*). Recent reports show that post-translational modifications of HMGB1, such as acetylation and phosphorylation, are associated with its translocation and release in inflammatory cells exposed to pathogen-related molecules including lipopolysaccharide (LPS) (*Bonaldi et al., 2003*; *Ito, Fukazawa & Yoshida, 2007*; *Youn & Shin, 2006*). The importance of extracellular HMGB1 in the inflammatory response has been demonstrated in inflammatory conditions; a neutralizing anti-HMGB1 antibody and HMGB1 antagonists attenuate cellular damage induced by inflammation (*Wang et al., 1999*; *Davé et al., 2009*). These reports indicate the importance of pathways or molecules that regulate HMGB1 release from activated inflammatory cells.

Sirtuin 1 (SIRT1) is a type III histone deacetylase that controls multiple genetic programs by acting on histone and non-histone substrates (*Xie, Zhang & Zhang, 2013*). This protein is a vital regulator of various physiological and metabolic processes such as energy metabolism (*Purushotham et al., 2009*), aging (*Tissenbaum & Guarente, 2001*), apoptosis (*Motta et al., 2004*), mitochondrial biogenesis (*Brenmoehl & Hoeflich, 2013*), and the stress response (*Brunet et al., 2004*). Recent studies also demonstrate that SIRT1 is directly involved in cellular inflammatory responses by deacetylating inflammation-related transcription factors such as nuclear factor-kappa B (NF-κB) and activator protein-1 (AP-1), which suppresses the transcription of diverse inflammation-responsive genes (*Feige & Auwerx, 2008*; *Zhang & Kraus, 2010*). Furthermore, we demonstrated that transcriptional upregulation of SIRT1 by peroxisome proliferator-activated receptor δ (PPARδ) and PPAR$\gamma$ inhibits HMGB1 release by decreasing its LPS-induced acetylation, indicating that SIRT1 deacetylates HMGB1 (*Hwang et al., 2012*, *2014*). While genetic ablation of SIRT1 increases the secretion and expression of proinflammatory cytokines, SIRT1 activators prevent the production of tumor necrosis factor-$\alpha$, monocyte chemoattractant protein-1, and interleukin (IL)-8 (*Dong et al., 2014*;

*Yang et al., 2007*), highlighting the central role of SIRT1 in the regulation of cellular inflammatory responses.

Formononetin, a herbal isoflavonoid, was isolated from the medicinal plant *Astragalus membranaceus* and has a variety of biological activities including anti-tumor (*Auyeung, Law & Ko, 2012*; *Chen et al., 2011*), wound healing (*Huh et al., 2011*), antioxidant (*Mu et al., 2009*), and anti-inflammatory (*Krenn & Paper, 2009*; *Lai et al., 2013*) effects. Specifically, formononetin inhibits inflammation-related gene expression by blocking the NF-κB and AP-1 signaling pathways in animal models of inflammatory diseases (*Chen et al., 2007*; *Hämäläinen et al., 2007*). In particular, synthetic derivatives of formononetin increase the activity of PPARδ, indicating this compound is useful to treat inflammation-related diseases (*Zhao et al., 2017*). Furthermore, we showed that activation of PPARδ and PPARγ by specific ligands induces SIRT1 expression in human coronary artery endothelial cells (*Kim et al., 2012*) and RAW264.7 cells (*Hwang et al., 2014*). Thus, we hypothesized that formononetin may modulate cellular inflammatory responses by inhibiting HMGB1 release via upregulation of SIRT1. Here, we show that formononetin reduces LPS-induced HMGB1 acetylation by upregulating SIRT1 in a PPARδ-dependent manner, thereby blocking HMGB1 release into the extracellular environment.

# MATERIALS AND METHODS

## Materials

Formononetin, actinomycin D (Act D), cycloheximide (CHX), Ponceau S solution, resveratrol, sirtinol, MTT, LPS (*Escherichia coli* 0111:B4), curcumin, genistein, and an anti-β-actin polyclonal antibody were obtained from Sigma-Aldrich Co. (St. Louis, MO, USA). GSK0660 and the luciferase assay system were purchased from Tocris Bioscience (Bristol, UK) and Promega (Madison, WI, USA), respectively. Monoclonal antibodies specific for HMGB1 and PPARδ were supplied by Epitomics (Burlingame, CA, USA). Monoclonal antibodies specific for acetyl-lysine, lamin B, and α-tubulin as well as a polyclonal antibody specific for SIRT1 were supplied by Santa Cruz Biotechnology (Dallas, TX, USA).

## Cell culture

Human primary peripheral blood macrophages and RAW264.7 murine macrophage-like cells were purchased from STEMCELL Technologies (Vancouver, BC, Canada) and the Korean Cell Line Bank (Seoul, Korea), respectively. Primary human macrophages and RAW264.7 cells were maintained in RPMI (Roswell Park Memorial Institute) 1640 and DMEM (Dulbecco's modified Eagle medium) containing antibiotics and 10% FCS at 37 °C in a 5% humidified $CO_2$ incubator, respectively.

## Cell viability assays

RAW264.7 cells were stimulated with 30 μM formononetin for the indicated duration or the indicated dose of formononetin for 24 h in 24-well plates. Thereafter, MTT assays and trypan blue exclusion were performed to determine the cell viability. For trypan blue exclusion, the collected cells were mixed with trypan blue solution (0.4%), and then viable

cells were determined by a hemocytometer. For MTT assay, the cells were incubated for final 2h in medium containing MTT solution (0.1 mg/ml). Following removing the medium, the absorbance at 570 nm was measured using formazan crystals solution dissolved in acidified isopropanol.

### Western blot analysis

Protein levels were analyzed by immunoblot as described previously (*Hwang et al., 2015*). Briefly, RAW264.7 cells washed with ice-cold PBS were lysed and aliquots of the resulting whole-cell lysates or conditioned media were analyzed by immunoblot with indicated antibodies. Immuno-reactive bands were detected using WesternBright ECL (Advansta Co., Menlo Park, CA, USA).

### Measurement of extracellular HMGB1

Levels of HMGB1 released into culture media were determined using a previously described method (*Hwang et al., 2012*). Briefly, the relative amounts of HMGB1 were determined in the conditioned media of RAW264.7 cells treated with the indicated reagents for the indicated durations. The 80% ice-cold acetone was used to precipitate the proteins in the conditioned media. After centrifugation, the pellets were obtained and washed with 80% ice-cold acetone. Following resuspension in SDS-PAGE sample buffer, the levels of HMGB1 released into culture media were analyzed by immunoblot.

### Fractionation of nuclear and cytoplasmic proteins

Cellular fractions were prepared using a previously described method (*Hwang et al., 2015*). Briefly, RAW264.7 cells were washed in PBS, suspended in lysis solution for 15 min at 4 °C to swell. Nonidet P-40 (final 0.1% concentration) was added to the lysates and then vortexed vigorously for 20 s. Following centrifugation (13,000*g*) for 20 s, the supernatant containing cytosolic fraction was obtained and the resulting pellet was lysed by a PRO-PREP Protein Extraction Solution. Following standing for 20 min on ice, the nuclear fraction (supernatant) was obtained by centrifugation.

### Reporter gene assay

The luciferase construct containing mouse SIRT1 promoter was a gift from Dr. Toren Finkel (NIH, MD, USA). The promoter activity of SIRT1 was measured as described previously (*Hwang et al., 2014*). Briefly, 1 μg of the SIRT1 luciferase reporter plasmid and 0.5 μg of pSV β-Gal (SV40 β-galactosidase expression vector) were introduced into RAW264.7 cells by SuperFect reagent (Qiagen, Valencia, CA, USA). After 38 h, the cells were treated with GSK0660 for 30 min prior to stimulation with formononetin for 24 h. Then, the cells were lysed by adding the luciferase reporter lysis buffer (Promega, Madison, WI, USA) and then aliquots of the lysates were used to determine luciferase activity.

### Small interfering RNA-mediated gene silencing

The indicated small interfering RNA (siRNA) was introduced into RAW264.7 cells in serum-containing medium using SuperFect (Qiagen, Valencia, CA, USA) as described previously (*Hwang et al., 2014*). Briefly, siRNA targeting scrambled non-specific

sequences (Ambion, Austin, TX, USA), PPARδ (Ambion, Austin, TX, USA), or SIRT1 designed against nucleotides (5′-TAATATCTGAGGCACTTCA-3′ and 5′-TGAAGTGCCTCAGATATTA-3′) of mouse (Bioneer, Daejeon, Korea) was introduced into the cells for 6 h. The cells were then cultured for further 38 h in fresh medium. At which time, the indicated reagents were added into the cells for the indicated durations. Gene silencing was analyzed by immunoblot.

### Real-time polymerase chain reaction

Levels of SIRT1 and HMGB1 mRNA were analyzed by real-time polymerase chain reaction (PCR) as described previously (*Hwang et al., 2014*). Briefly, total RNA was converted into cDNA by a reverse transcription kit (TOPscript RT DryMIX; Enzynomics, Seoul, Korea). Real-time PCR was carried out using equal amount of cDNA in a 20 μl reaction solution containing primers and 1 × SYBR PCR mix (Takara Bio Inc., Otsu, Japan). The PCR condition: initial denaturation at 94 °C for 20 min, followed by 42 cycles of 25 s at 95 °C, 44 s at 58.2 °C, and 40 s at 72 °C. The primers were as follows: SIRT1, 5′-AGAACCACCAAAGCGGAAA-3′ and 5′-TCCCACAGGAGACAGAAACC-3′; HMGB1, 5′-TACCGCCCCAAAATCAAAGG-3′ and 5′-TCTCATAGGGCTGCTTGTCA-3′; and GAPDH, 5′-CATGGCCTTCCGTGTTCCTA-3′ and 5′-CCTGCTTCACCACCTTCTTGAT-3′.

### Co-immunoprecipitation

Immunoprecipitation was performed using a previously described method (*Hwang et al., 2015*). Briefly, the protein G Sepharose was added to whole-cell lysates to pre-clear and then the pre-cleared lysates were mixed with 1 μg of an anti-HMGB1 antibody. After incubation overnight at 4 °C, the mixture was reacted with protein G Sepharose for 4 h. Mixtures were extensively washed with PBS and then boiled in gel-loading buffer. The immunoblot analysis was performed using an anti-acetyl-lysine antibody (Santa Cruz Biotechnology, Dallas, TX, USA).

### Statistical analysis

Data are expressed as mean ± standard error (SE). The significance in statistical analysis was evaluated by a one-way ANOVA, followed by Tukey–Kramer test. A value of $p < 0.05$ was considered statistically significant.

## RESULTS

### Formononetin inhibits LPS-induced release of HMGB1 in both murine and human macrophages

To determine the optimal concentration of formononetin, we determined the viability of RAW264.7 cells treated with various concentrations of formononetin for 24 h or with 30 μM formononetin for various durations. Treatment with concentrations of formononetin up to 30 μM did not elicit cytotoxic effects on RAW264.7 cells, and cell viability remained high following treatment with 30 μM formononetin for up to five days (Figs. 1A and 1B). Thus, we selected 30 μM formononetin as the optimal concentration for subsequent experiments using RAW264.7 cells.

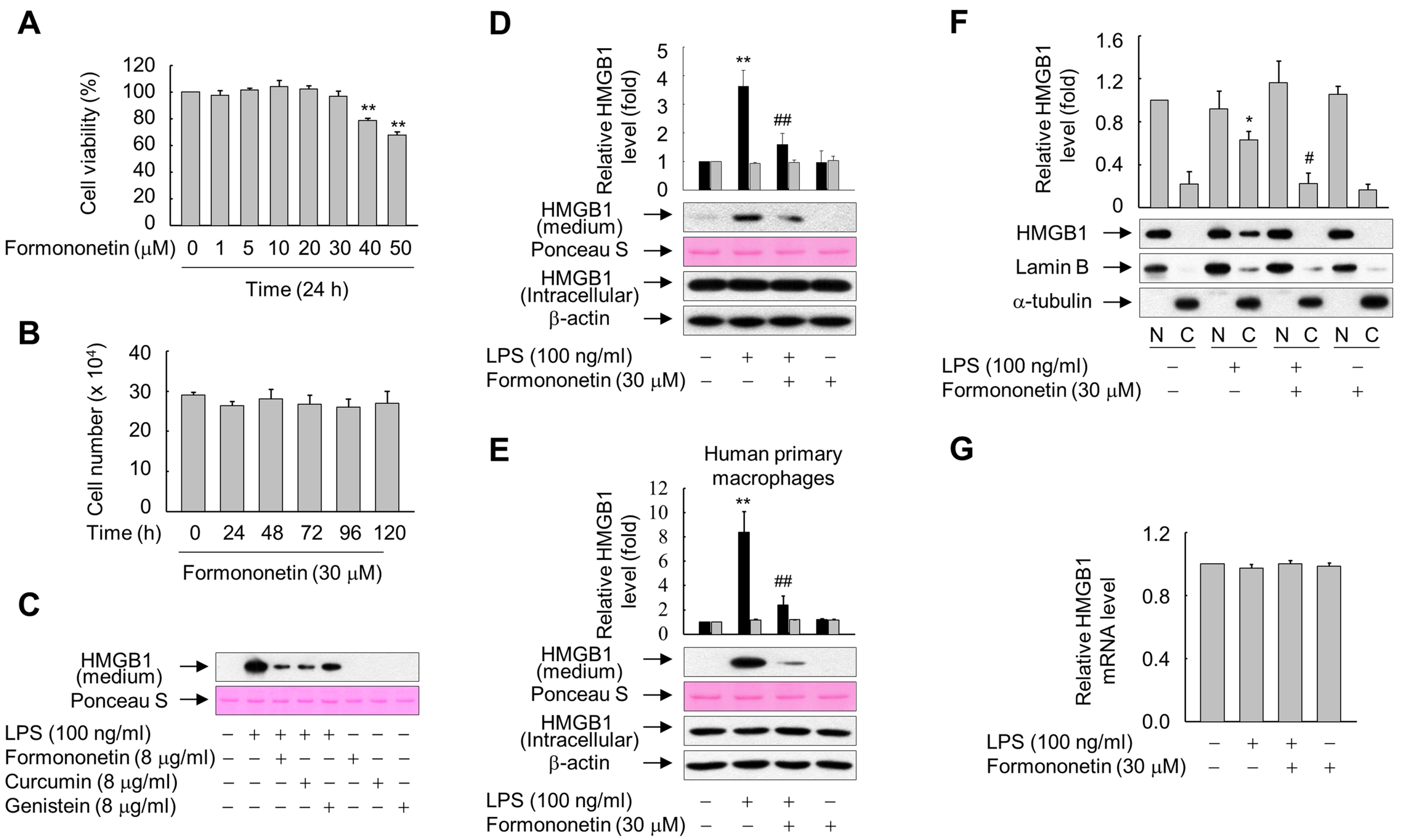

**Figure 1 Effects of formononetin on the LPS-induced release and translocation of HMGB1.** (A, B) RAW264.7 cells cultured in serum-free medium for 16 h were treated with the indicated concentrations of formononetin for 24 h (A) or with 30 mM formononetin for the indicated durations (B). Cell viability was determined by the MTT (A) and trypan blue exclusion (B) assays. (C) RAW264.7 cells maintained in serum-free medium for 16 h were stimulated with LPS in the presence or absence of indicated herbal compound for 24 h. Equal volumes of conditioned media were analyzed by immunoblotting. Ponceau S staining was used as the loading controls. (D, E) RAW264.7 cells (D) or human primary macrophages (E) cultured in serum-free medium for 16 h were stimulated with LPS in the presence or absence of formononetin for 24 h. Equal volumes of conditioned media or aliquots of whole-cell lysates were analyzed by immunoblotting. Ponceau S staining and β-actin were used as the loading controls. Black and gray bars indicate secreted HMGB1 and intracellular HMGB1, respectively. (F) RAW264.7 cells treated with LPS in the presence or absence of formononetin for 24 h were fractionated into nuclear (N) and cytosolic (C) fractions. The localization of HMGB1 was determined by Western blot analysis with the indicated antibodies. (G) RAW264.7 cells were treated with LPS in the presence or absence of formononetin. Following incubation for 24 h, total RNA was isolated and the levels of SIRT1 mRNA were analyzed by real-time PCR. The results are plotted as the mean ± SE ($n = 3$ or 4). $^{*}p < 0.05$, $^{**}p < 0.01$ compared with the untreated group; $^{\#}p < 0.05$, $^{\#\#}p < 0.01$ compared with the LPS-treated group.

Because many herbal compounds including curcumin and genistein reported to exhibit anti-inflammatory activity (*Biswas & Rahman, 2008*; *Dharmappa et al., 2010*), we compared the effect of formononetin on the HMGB1 release with curcumin and genistein in RAW264.7 cells treated with LPS. The level of HMGB1 released into culture media was increased in RAW264.7 cells exposed to LPS, and this increase was markedly reduced in the presence of herbal compounds. In particular, the effects of formononetin and curcumin were superior to those of genistein (Fig. 1C). By contrast, neither LPS nor formononetin affected the expression level of endogenous HMGB1 (Fig. 1D). Similar results were obtained from the human primary macrophages, indicating

that formononetin affects LPS-induced HMGB1 release, but not HMGB1 expression, in both murine and human macrophages (Fig. 1E).

HMGB1 is reported to translocate from the nucleus into the cytoplasm in response to inflammatory signals such as LPS (*Bonaldi et al., 2003*; *Youn & Shin, 2006*). Therefore, we examined whether formononetin affects this translocation of HMGB1 in LPS-stimulated RAW264.7 cells. While translocation of HMGB1 into the cytoplasm was increased in cells exposed to LPS, this was significantly suppressed by formononetin (Fig. 1F). However, the expression of HMGB1 mRNA was not affected by formononetin in cells treated with or without LPS (Fig. 1G). These results suggest that formononetin decreases the release of HMGB1 by inhibiting its translocation rather than expression in LPS-primed RAW264.7 cells.

## Formononetin upregulates SIRT1 expression in RAW264.7 cells

Formononetin increased protein expression of SIRT1 in RAW264.7 cells in a concentration- and time-dependent manner. SIRT1 protein expression was significantly increased in cells treated with 20–30 μM formononetin for 24 h (Fig. 2A) and peaked at 12–24 h in cells treated with 30 μM formononetin (Fig. 2B). Similarly, the mRNA level of SIRT1 was time-dependently upregulated by formononetin (Fig. 2C). In addition, the inhibitory effect of formononetin on the LPS-stimulated release of HMGB1 was significant at 6 h pre-treatment and the maximal inhibitory effect of formononetin was observed with a pre-treatment of 24 h which corresponds to the time of maximal induction of SIRT1 expression upon formononetin treatment (Fig. 2D).

To elucidate the mechanisms by which formononetin induces SIRT1 expression, we determined the effects of Act D (a RNA synthesis inhibitor) and CHX (a protein synthesis inhibitor). While formononetin significantly increased mRNA expression of SIRT1, this was significantly reduced in the presence of Act D or CHX (Fig. 2E). These results indicate that de novo synthesis of mRNA as well as of proteins that act on the *SIRT1* gene promoter is indispensable for the induction of *SIRT1* mRNA by formononetin in RAW264.7 cells.

## Formononetin induces SIRT1 expression via PPARδ in RAW264.7 cells

To further examine the mechanisms by which formononetin upregulates SIRT1 expression, we evaluated the role of PPARδ, a nuclear receptor that regulates the transcription of a variety of target genes (*Kidani & Bensinger, 2012*; *Mangelsdorf et al., 1995*), by transfecting RAW264.7 cells with siRNA against PPARδ. The protein level of PPARδ was reduced in cells transfected with PPARδ-targeting siRNA, but not in cells transfected with control siRNA composed of a pool of nonspecific sequences (Fig. 2F). Transfection of PPARδ-targeting siRNA attenuated the induction of SIRT1 expression by formononetin, whereas transfection of control siRNA did not (Fig. 2G). In line with these findings, GSK0660, a specific inhibitor of PPARδ, significantly attenuated the formononetin-induced increase in SIRT1 promoter activity (Fig. 2H). These results

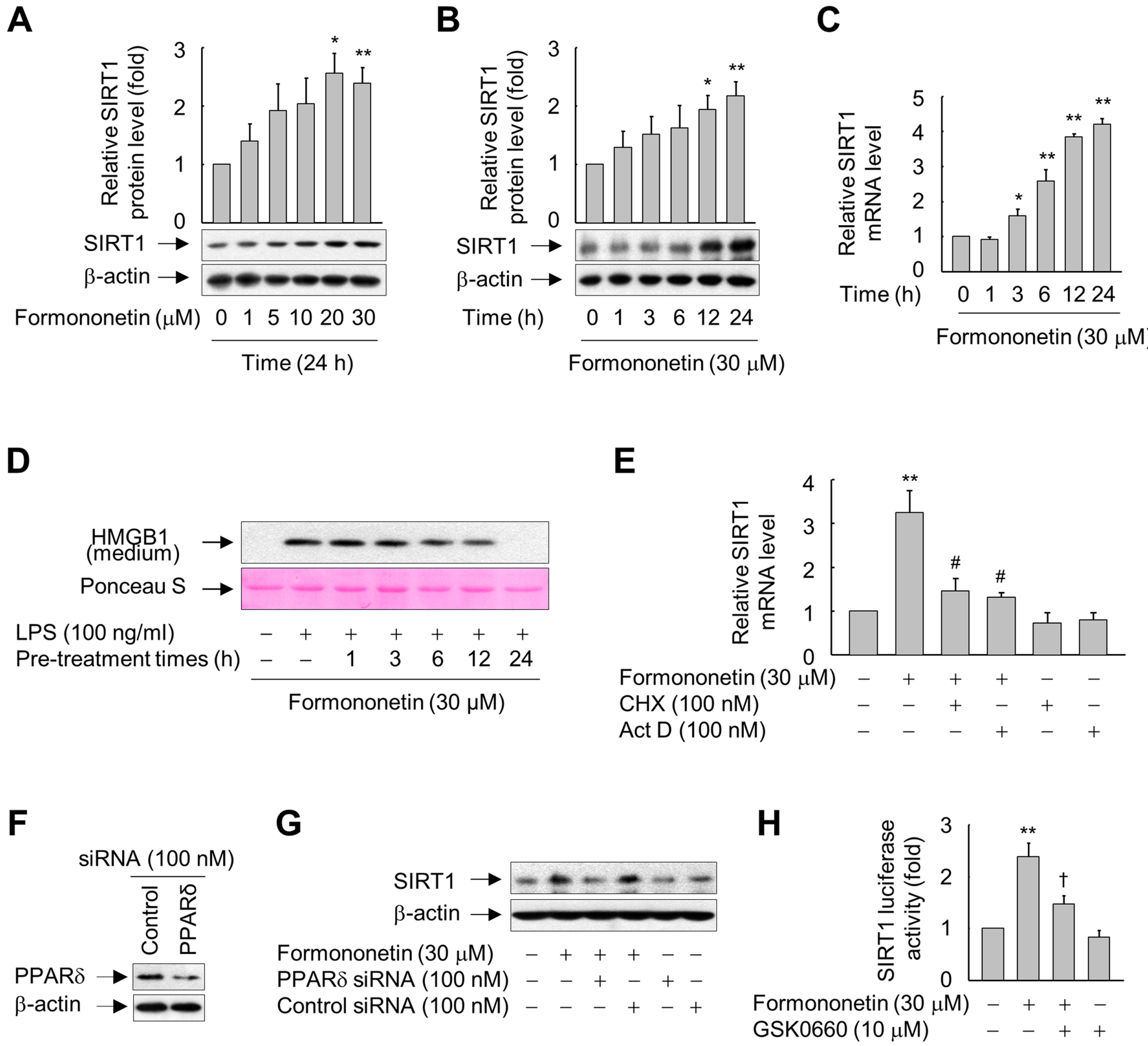

**Figure 2 Involvement of PPARδ in formononetin-mediated upregulation of SIRT1 in RAW264.7 cells.** (A, B) Cells cultured in serum-free medium for 16 h were incubated with various concentrations of formononetin for 24 h (A) or with 30 mM formononetin for the indicated durations (B). Aliquots of whole-cell lysates were analyzed by immunoblotting. Representative blots are provided. Fold changes in the SIRT1/β-actin ratio relative to that in the untreated group are shown as mean ± SE ($n = 3$). (C) Cells cultured in serum-free medium for 16 h were stimulated with formononetin for the indicated durations. After incubation for 24 h, total RNA was isolated and the levels of SIRT1 mRNA were analyzed by real-time PCR. The results are expressed as the mean ± SE ($n = 3$). (D) Cells maintained in serum-free medium for 16 h were pre-treated with formononetin for indicated times. Following washing with fresh medium, the cells were stimulated with LPS for 24 h. Equal volumes of conditioned media were analyzed by immunoblotting. Ponceau S staining was used as the loading controls. (E) Cells cultured in serum-free medium for 16 h were incubated with CHX or Act D in the presence or absence of formononetin. After incubation for 24 h, total RNA was isolated and the levels of SIRT1 mRNA were analyzed by real-time PCR. The results are expressed as the mean ± SE ($n = 3$). (F) Cells were transfected with siRNA against PPARδ or control and grown for 38 h. The cells were then lysed and aliquots of whole-cell lysates were subjected to Western blot analysis. (G) Cells transfected with PPARδ-targeting or control siRNA for 38 h were stimulated with formononetin for 24 h. Aliquots of whole-cell lysates were analyzed by immunoblotting. (H) Cells transfected with 1 μg of the SIRT1 luciferase reporter plasmid and 0.5 μg of pSV b-Gal for 38 h were pretreated with GSK0660 for 30 min and then exposed to formononetin for 24 h. Luciferase activity was normalized to β-galactosidase activity. The results are expressed as the mean ± SE ($n = 3$). $^{*}p < 0.05$, $^{**}p < 0.01$ compared with the untreated group; $^{\#}p < 0.05$ compared with the LPS-treated group; $^{\dagger}p < 0.05$ compared with the formononetin-treated group.

suggest that formononetin upregulates SIRT1 expression via PPARδ at the transcriptional level.

## SIRT1 is essential for inhibition of LPS-induced HMGB1 release by formononetin

To investigate the direct effect of SIRT1 on LPS-induced HMGB1 release, we examined the levels of SIRT1 protein and released HMGB1 in RAW264.7 cells exposed to LPS in the presence or absence of formononetin. A high level of HMGB1 was released upon LPS treatment, whereas this was reduced in the presence of formononetin. On the other hand, the level of SIRT1 protein was significantly suppressed in LPS-treated RAW264.7 cells. However, this LPS-mediated repression of SIRT1 was recovered in the presence of formononetin, indicating that SIRT1 is critical for modulation of LPS-induced HMGB1 release by formononetin (Fig. 3A).

To further clarify the functional significance of formononetin-mediated upregulation of SIRT1 in RAW264.7 cells, we manipulated the expression and activity of SIRT1 using siRNA or chemicals. The levels of SIRT1 protein were diminished in cells transfected with SIRT1 siRNA, however control siRNAs had no effect on the levels of either protein (Fig. 3B). Transfection of SIRT1-targeting siRNA significantly attenuated the inhibitory effect of formononetin on LPS-induced HMGB1 release (Fig. 3C). Consistently, inhibition of SIRT1 activity by sirtinol also prevented inhibition of HMGB1 release by formononetin (Fig. 3D). By contrast, activation of SIRT1 by resveratrol inhibited LPS-induced HMGB1 release. Furthermore, resveratrol treatment potentiated the inhibitory effects of formononetin, suggesting that SIRT1 plays a role in the suppression of HMGB1 release by formononetin (Fig. 3E). These results indicate that formononetin inhibits LPS-induced HMGB1 release by regulating SIRT1 expression.

## SIRT1-mediated deacetylation of HMGB1 underlies the inhibition of its release by formononetin

Inflammatory signal-mediated acetylation of HMGB1 is critical for its release into the extracellular compartment and acetylated HMGB1 is a substrate of SIRT1 (*Bonaldi et al., 2003*; *Hwang et al., 2014*; *Rickenbacher et al., 2014*); therefore, we evaluated whether formononetin affects LPS-induced acetylation of HMGB1. When RAW264.7 cells were stimulated with LPS for 6 h, the level of acetylated HMGB1 in an immunoprecipitate obtained using an anti-HMGB1 antibody was significantly enhanced. However, formononetin reduced this increase in acetylated HMGB1 in a concentration-dependent manner, indicating that formononetin is involved in the deacetylation of HMGB1 primed by LPS (Fig. 4A).

To evaluate whether this inhibition of LPS-induced HMGB1 acetylation by formononetin correlates with the level of SIRT1 in RAW264.7 cells, we knocked down SIRT1. Transfection of SIRT1-targeting siRNA significantly prevented the decrease in acetylated HMGB1 by formononetin in LPS-exposed RAW264.7 cells (Fig. 4B). Transfection of PPARδ-targeting siRNA elicited the same effect (Fig. 4C). These results indicate that formononetin reduces HMGB1 acetylation via PPARδ-mediated

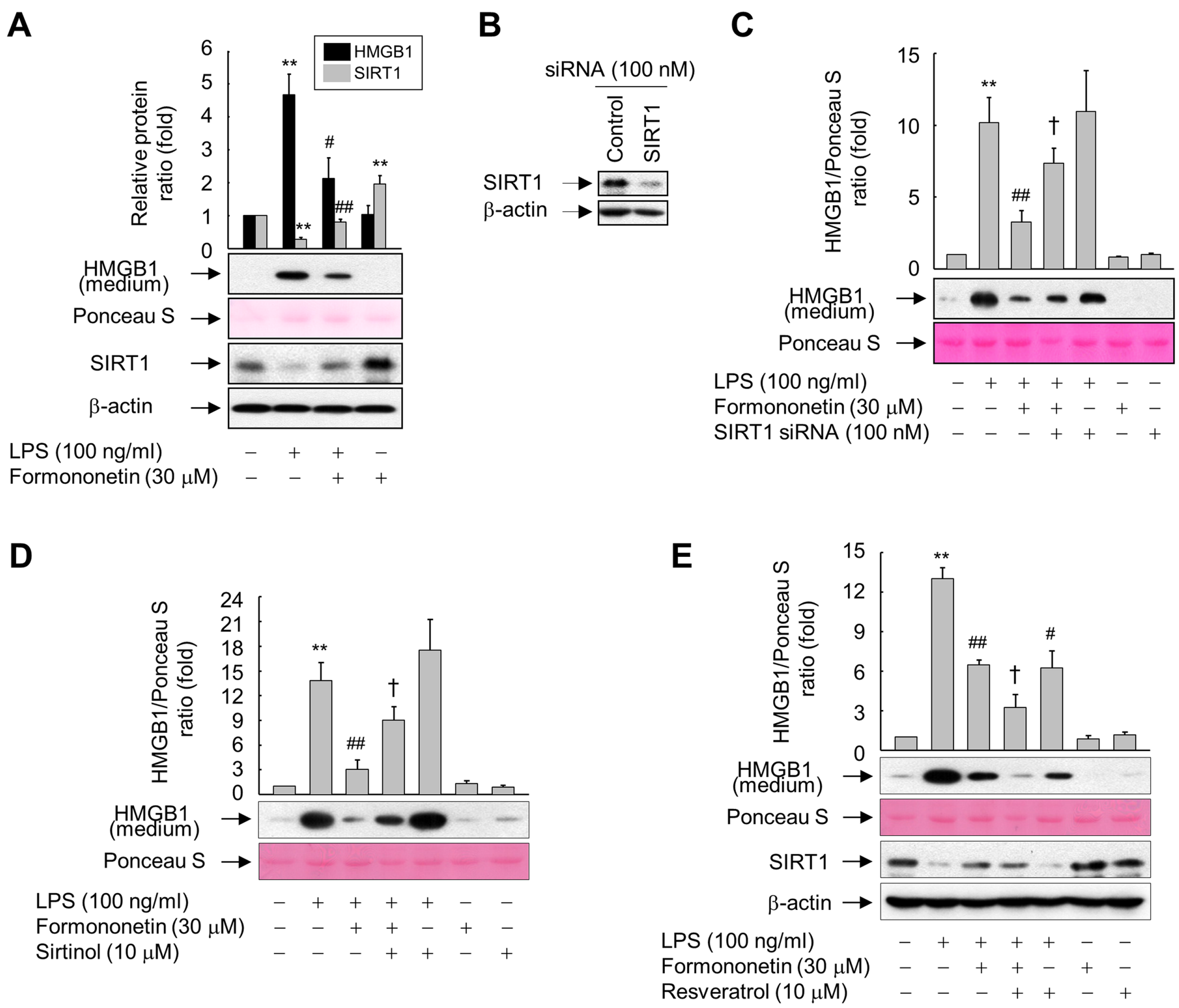

**Figure 3 Involvement of SIRT1 in the formononetin-mediated inhibition of LPS-induced HMGB1 release in RAW264.7 cells.** (A) Cells cultured in serum-free medium for 16 h were stimulated with LPS in the presence or absence of formononetin for 24 h. (B) Cells were transfected with siRNA against SIRT1 or control and incubated for 38 h. (C) Cells transfected with or without SIRT1-targeting siRNA for 38 h were exposed to LPS in the presence or absence of formononetin for 24 h. (D, E) Cells pretreated with sirtinol (D) or resveratrol (E) for 30 min were stimulated with LPS in the presence or absence of formononetin for 24 h. Equal volumes of conditioned media or aliquots of whole-cell lysates were analyzed by immunoblotting with the indicated antibodies. Ponceau S staining and β-actin were used as a loading control. Representative blots are provided. The fold changes in the SIRT1/β-actin or HMGB1/Ponceau S ratio relative to that in the untreated group are shown as mean ± SE ($n = 3$). $^{**}p < 0.01$ compared with the untreated group; $^{\#}p < 0.05$, $^{\#\#}p < 0.01$ compared with the LPS-treated group; $^{\dagger}p < 0.05$ compared with the LPS plus formononetin-treated group.

upregulation of SIRT1, thereby inhibiting the release of HMGB1 into the extracellular milieu.

## DISCUSSION

HMGB1 plays physiological and pathological roles by acting as an intracellular structural protein and an extracellular cytokine (*Ueda & Yoshida, 2010*; *Andersson & Tracey, 2011*). Although the roles of extracellular HMGB1 in the pathogenesis of inflammatory disease are well established, the regulatory mechanisms underlying HMGB1 release or therapeutic agents that can impede its release was not fully elucidated. Here, we showed

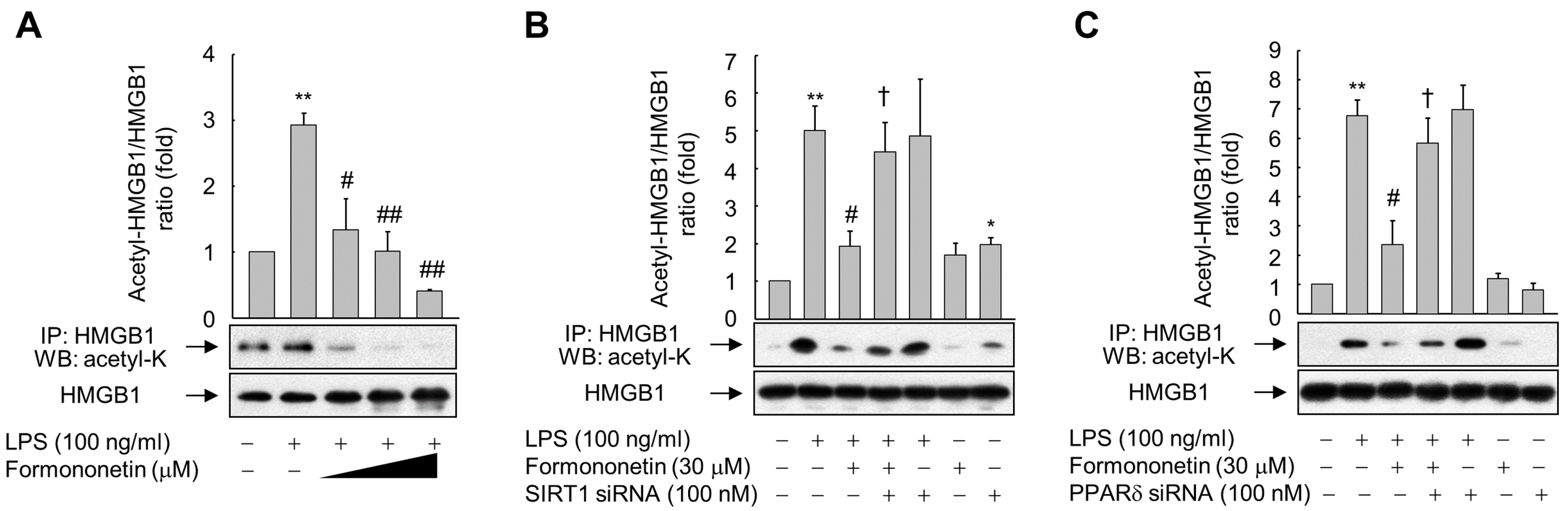

**Figure 4 Effect of formononetin on LPS-induced HMGB1 acetylation.** (A) RAW264.7 cells cultured in serum-free medium for 16 h were stimulated with LPS in the presence of increasing concentrations (10, 20, and 30 mM) of formononetin for 6 h. (B, C) Cells transfected with SIRT1-targeting siRNA (B) or PPARδ-targeting siRNA (C) for 38 h were exposed to LPS in the presence or absence of formononetin for 6 h. Whole-cell lysates were immunoprecipitated with an anti-HMGB1 antibody, and then acetylated HMGB1 was detected by immunoblot analysis with an anti-acetyl-lysine antibody. Representative blots are provided. The fold changes in the acetylated HMGB1/total HMGB1 ratio relative to that in the untreated group are shown as mean ± SE ($n = 3$). $^{**}p < 0.01$ compared with the untreated group; $^{\#}p < 0.05$, $^{\#\#}p < 0.01$ compared with the LPS-treated group; $^{\dagger}p < 0.05$ compared with the LPS plus formononetin-treated group.

that formononetin inhibited LPS-induced release of HMGB1 in RAW264.7 cells. This inhibition was mediated by PPARδ-dependent upregulation of SIRT1, a class III deacetylase involved in cellular inflammatory responses (*Brunet et al., 2004*; *Yeung et al., 2004*; *Zhang et al., 2010*). SIRT1 expression was also upregulated at the transcriptional level in RAW264.7 cells treated with formononetin. Modulation of SIRT1 expression and activity by siRNAs and chemicals abolished the inhibitory effect of formononetin on HMGB1 release. In addition, SIRT1 upregulated by formononetin deacetylated HMGB1, which inhibited release of HMGB1. This demonstrates that formononetin has anti-inflammatory actions in LPS-stimulated RAW264.7 cells. These results are in line with the previous finding that formononetin elicits anti-inflammatory effects by upregulating PPARγ expression in an animal model of LPS-induced acute lung injury (*Ma et al., 2013*). In addition, formononetin attenuates hydrogen peroxide- and IL-1β-induced activation of NF-κB in retinal ganglion cells and the insulinoma cell line INS-1, respectively (*Jia et al., 2014*; *Wang et al., 2012*). Although the molecular mechanisms underlying formononetin-mediated anti-inflammatory responses have not been fully elucidated until now, the present study clearly demonstrated that formononetin inhibits LPS-induced release of HMGB1 in the mouse macrophage cell line RAW264.7, suggesting that formononetin is a promising therapeutic agent for inflammation-related disorders.

The release of HMGB1 during inflammatory responses is closely linked with its post-translational modifications such as acetylation and phosphorylation (*Bonaldi et al., 2003*; *Youn & Shin, 2006*). Consistent with previous studies, formononetin inhibited LPS-induced acetylation of HMGB1, leading to suppression of its release. This effect of formononetin on HMGB1 release was intimately correlated with the level of SIRT1

expression, indicating that SIRT1 deacetylates HMGB1 and thereby inhibits its release. This result is in line with previous reports indicating that SIRT1 deacetylates inflammation-related transcription factors such as AP-1 and NF-κB, and thereby modulates the progression of inflammation by suppressing the transcription of diverse inflammation-related genes (*Yang et al., 2007*; *Yeung et al., 2004*; *Zhang et al., 2010*). These results provide a rationale for the use of SIRT1 activators as therapeutic agents in inflammatory diseases as shown previous studies using resveratrol to activate the SIRT1 (*Xu et al., 2014*; *Dong et al., 2015*). In fact, a recent study demonstrated that inflammatory diseases are closely associated with a reduced SIRT1 protein level (*Xie, Zhang & Zhang, 2013*). Because release of HMGB1 is intimately correlated with its post-translational modifications along with decreased SIRT1 expression, it may be possible to suppress inflammatory reactions by inducing SIRT1 expression using formononetin.

Formononetin-mediated upregulation of SIRT1 was critical for inhibition of LPS-induced HMGB1 release. SIRT1, a $NAD^+$-dependent deacetylase, is implicated in diverse cellular processes, such as stress responses, aging, energy metabolism, and inflammation, through its deacetylase activity (*Brunet et al., 2004*; *Chen et al., 2005a*; *Cohen et al., 2004*; *Feige & Auwerx, 2008*; *Yeung et al., 2004*; *Zhang & Kraus, 2010*; *Zhang et al., 2010*). Although transcriptional regulation of SIRT1 in mammalian cells has been mainly established in the context of energy metabolism-related pathways such as caloric restriction (*Chen et al., 2005a*; *Cohen et al., 2004*), transcription factors, including TLX, BRCA1, HIC1, and E2F1, are also implicated in the regulation of SIRT1 expression (*Chen et al., 2005b*; *Iwahara et al., 2009*; *Wang et al., 2006*, *2008*). However, the transcriptional regulation of SIRT1 is complex and the underlying mechanism is unclear. The nuclear hormone receptor PPARδ was recently demonstrated to regulate SIRT1 expression in various cell lineages (*Kim et al., 2012*; *Okazaki et al., 2010*). PPARδ was initially shown to promote SIRT1 expression in human hepatocytes via an unconventional mechanism in which specificity protein 1 plays a central role, rather than the PPAR-response element (*Okazaki et al., 2010*). PPARδ activation also induces SIRT1 expression in vascular endothelial cells (*Kim et al., 2012*). On the other hand, formononetin, a compound extracted from *Sophora flavescens* roots, significantly increases PPARδ activity in a concentration-dependent manner (*Quang et al., 2013*), indicating that transactivation of PPARδ by formononetin is linked to SIRT1 expression. This result is in line with our finding that formononetin-induced SIRT1 expression in a PPARδ-dependent manner.

## CONCLUSION

To our knowledge, this is the first report to show that formononetin inhibits HMGB1 release by upregulating SIRT1 transcription and thus inducing HMGB1 deacetylation in LPS-treated RAW264.7 cells. This novel finding has important implications for our understanding of the molecular mechanism underlying the transcriptional regulation of SIRT1 as well as the anti-inflammatory effect of formononetin. In light of these observations, formononetin-mediated enhancement of SIRT1 activity in macrophages is likely a new therapeutic strategy for inflammatory disorders.

### Funding

This paper was supported by the KU Research Professor Program of Konkuk University. The funders had no role in study design, data collection and analysis, decision to publish, or preparation of the manuscript.

### Grant Disclosures

The following grant information was disclosed by the authors:
KU Research Professor Program of Konkuk University.

### Competing Interests

The authors declare that they have no competing interests.

### Author Contributions

- Jung Seok Hwang conceived and designed the experiments, performed the experiments, analyzed the data, contributed reagents/materials/analysis tools, wrote the paper, prepared figures and/or tables.
- Eun Sil Kang performed the experiments, analyzed the data, contributed reagents/materials/analysis tools, prepared figures and/or tables.
- Sung Gu Han analyzed the data, contributed reagents/materials/analysis tools.
- Dae-Seog Lim analyzed the data, contributed reagents/materials/analysis tools.
- Kyung Shin Paek contributed reagents/materials/analysis tools, reviewed drafts of the paper.
- Chi-Ho Lee wrote the paper, reviewed drafts of the paper.
- Han Geuk Seo conceived and designed the experiments, wrote the paper, prepared figures and/or tables, reviewed drafts of the paper.

### Data Availability

The raw data has been supplied as Supplemental Dataset Files.

### Supplemental Information

Supplemental information for this article can be found online at http://dx.doi.org/10.7717/peerj.4208#supplemental-information.

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
