# Peer review of "Formononetin inhibits lipopolysaccharide-induced release of high mobility group box 1 by upregulating SIRT1 in a PPARδ-dependent manner"

_PeerJ, doi:10.7717/peerj.4208_

## Round 0.1 · original submission · Major Revisions

· Academic Editor

Major Revisions

Please answer to all of the issues raised by both reviewers. In particular, reviewer 1 suggests to use primary macrophages to confirm results.

Reviewer 1 ·

Basic reporting

The article is clear and english is correct.
Results of figure 2 should be inserted in figure 1 as panels C and D. Similarly, data of figure 4 should be inserted in figure 3 as panels C to F. Data presented in the supplemental figures should be inserted in the main figures.
Some references should be added in particular on the effect of resveratrol on HMGB1 release (e.g. Xu et al. Shock 2014; Dong et al. Free Radic Biol. Med 2015).

Experimental design

The objective of the study is to assess if formononetin modulates HMGB1 secretion in response to LPS. Overall, the methods are well described, the experiments are well designed with appropriate controls and technical replicates.
The authors should provide the quantification of the western-blots in figure 2A-B. In figure 2B, it seems that the total amount of HMGB1 is increased with LPS or formononetin treatment. The authors should analyze HMGB1 mRNA level upon formononetin treatment.
RAW264.7 cell line is not the most suitable model to study HMGB1 release and the effect of formononetin on HMGB1 secretion should be confirmed in human primary macrophages.

Validity of the findings

The novelty of the study is rather limited as a large number of studies already demonstrated that many herbal compounds inhibit HMGB1 release by preventing its acetylation. It might be interesting to focus on the advantage of using formononetin compared to other similar compounds. Hence, it might be useful to compare the effect of formononetin on HMGB1 release with other similar compounds.
In figure 5A, treatment with formononetin alone significantly increased the level of SIRT1 and in figure 1B it seems that formononetin alone decreases the level of HMGB1 in the cytoplasm. It might be interesting to perform a pre-treatment with formononetin and then wash the cells and activate them with LPS in order to evaluate if the effect of formononetin treatment persists in time.
From authors conclusions, formononetin and resveratrol appear to modulate HMGB1 release via SIRT1 expression. Hence, why did they observe a synergic effect in figure 5D? If these two compounds modulate HMGB1 release via the same mechanism, no synergic effect should be observed. The authors should also analyze the level of SIRT1 in this experiment.

Reviewer 2 ·

Basic reporting

The paper deals with the role of formononetin as anti-inflammatory molecule. Basically, formononetin acts by inhibiting HMGB1 release via Sirt1 up-regulation. English form is correct and appropriate. References are accurate. Line 165: Please fix “. At which time”

Experimental design

The study is well designed. The results are well presented and understandable. Methods are well described with the exception of statistical analysis, which should be explained in a wider way.

Validity of the findings

Authors provide a functional explanation for HMGB1 release inhibition by formononetin. All experiments are well showed, results are consistent and statistically sounding. Conclusions are well written and results support the conclusions.

---

## Round 0.2 · Minor Revisions

· Academic Editor

Minor Revisions

Please, organise your figures 1 and 2 as suggested by reviewer 1.

Reviewer 1 ·

Basic reporting

The results of the two reference figures should be inserted in the manuscript (as figure 1C for reference figure 1 and as figure 2D for reference figure 2). In addition, it would be useful to highlight in the text that the maximal inhibitory effect of formononetin is observed with a pre-treatment of 24 hours which corresponds to the time of maximal induction of SIRT1 expression upon formononetin treatment (Fig.2B and C).

Experimental design

no comment

Validity of the findings

no comment

Additional comments

The authors replied to all my comments.
The results of the two reference figures are convincing and should be inserted in the manuscript (as figure 1C for reference figure 1 and as figure 2D for reference figure 2). In addition, it would be useful to highlight in the text that the maximal inhibitory effect of formononetin is observed with a pre-treatment of 24 hours which corresponds to the time of maximal induction of SIRT1 expression upon formononetin treatment (Fig.2B and C).

Reviewer 2 ·

Basic reporting

The authors have reviewed the paper and fixed where requested. Indeed, English form is now correct and references accurate.

Experimental design

Statistical analysis section has been enriched and it is now well understandable.

Validity of the findings

The paper is clear and results are well written. Experimental section is accurate and well designed. Conclusions are consistent.

---

## Round 0.3 · accepted · Accept

· Academic Editor

Accept

The authors have satisfactorily addressed the issues raised by the reviewers.